# LEF1 Enhances the Progression of Colonic Adenocarcinoma via Remodeling the Cell Motility Associated Structures

**DOI:** 10.3390/ijms221910870

**Published:** 2021-10-08

**Authors:** Li Xiao, Caixia Zhang, Xinyao Li, Chenshuang Jia, Lirong Chen, Yue Yuan, Qian Gao, Zheng Lu, Yang Feng, Ruixia Zhao, Xuewei Zhao, Sinan Cheng, Zhan Shu, Jie Xu, Wei Duan, Guochao Nie, Yingchun Hou

**Affiliations:** 1College of Life Sciences, Shaanxi Normal University, Xi’an 710119, China; xiaoli@snnu.edu.cn (L.X.); zhangcaixia@snnu.edu.cn (C.Z.); xiaxia1991@snnu.edu.cn (X.L.); jcs@snnu.edu.cn (C.J.); chenlirong@snnu.edu.cn (L.C.); yuan-yue@snnu.edu.cn (Y.Y.); gaoqian01@snnu.edu.cn (Q.G.); lzadzw@snnu.edu.cn (Z.L.); 15555328533@163.com (Y.F.); 15176837898@163.com (R.Z.); zxw_818@163.com (X.Z.); chengsinan@snnu.edu.cn (S.C.); sz10111529@snnu.edu.cn (Z.S.); xujie19@snnu.edu.cn (J.X.); 2School of Medicine, Deakin University, Geelong, VIC 3216, Australia; wei.duan@deakin.edu.au; 3Ukraine Joint Research Center for Nano Carbon Black, Yulin 537000, China; 4Optoelectronic Information Research Center, School of Physics and Telecommunication Engineering, Yulin Normal University, Yulin 537000, China; 5Guangxi Key Laboratory of Agricultural Resource Chemistry and Biotechnology, Yulin 537000, China

**Keywords:** LEF1, colonic cancer, cancer malignancy, cell microstructures, EMT

## Abstract

Lymphoid enhancer-binding factor 1 (LEF1) is a key transcription factor mediating the Wnt signaling pathway. LEF1 is a regulator that is closely associated with tumor malignancy and is usually upregulated in cancers, including colonic adenocarcinoma. The underlying molecular mechanisms of LEF1 regulation for colonic adenocarcinoma progression remain unknown. To explore it, the LEF1 expression in caco2 cells was inhibited using an shRNA approach. The results showed that downregulation of LEF1 inhibited the malignancy and motility associated microstructures, such as polymerization of F-actin, β-tubulin, and Lamin B1 in caco2 cells. LEF1 inhibition suppressed the expression of epithelial/endothelial-mesenchymal transition (EMT) relevant genes. Overall, the current results demonstrated that LEF1 plays a pivotal role in maintaining the malignancy of colonic adenocarcinoma by remodeling motility correlated microstructures and suppressing the expression of EMT-relevant genes. Our study provided evidence of the roles LEF1 played in colonic adenocarcinoma progression, and suggest LEF1 as a potential target for colonic adenocarcinoma therapy.

## 1. Introduction

Colonic adenocarcinoma is one of the leading causes of cancer-associated mortality from the digestive system [1,2]. Due to improved diagnostic and therapeutic strategies, as well as decreased societal risk factor exposure, the incidence rate of colonic adenocarcinoma has been steadily decreasing. However, the survival rate for colonic adenocarcinoma remains unsatisfactory [3]. The oncogenesis and progression of cancer involves multiple genetic mutations. The molecular mechanisms and key regulators of colonic adenocarcinoma progression and metastasis remain elusive and need to be investigated. Further exploration of the molecular mechanisms is essential to improve current diagnostic and treatment strategies for colonic adenocarcinoma.

Lymphoid enhancer-binding factor 1 (LEF1) is a transcription factor primarily involved in the canonical Wnt/β-catenin signaling pathway and is implicated in tumorigenesis and progression of multiple neoplasms [4,5]. LEF1 belongs to the T cell Factor (TCF)/LEF family of transcription factors [6], containing a highly conserved high mobility group (HMG) DNA-binding domain, and plays the role of nuclear effector in the Wnt/β catenin signaling pathway [6,7]. In the absence of Wnt signaling, LEF1 is bound to Groucho-related co-repressors, thus negatively regulating the expression of Wnt signaling genes [8]. Upon stabilization from Wnt signals, β-catenin displaces the Groucho-related co-repressors and promotes LEF1 transcription factor activity [5]. LEF1 is usually expressed in pre-B and T cells [9,10] and plays an important role in embryogenesis and cancer development [11]. When expressed in normal cells, the LEF1 locus produces two protein isoforms from two different RNA polymerase II promoters [12], which is a multifunctional protein that influences numerous cellular functions, such as regulation of Wnt signaling for cell proliferation, apoptosis, motility, and gene transcription [13,14]. Abnormal expression of LEF1 has been correlated with various tumors including colonic adenocarcinoma [15]. Particularly, while the deficiency of its own transcriptional activation potential in cells, LEF1 can mediate the expression of Wnt signaling genes via recruitment of β-catenin to the promoter of the target genes. Numerous downstream target genes could be affected, such as c-myc, cyclin-D1, survin, and VEGF [16,17].

Reorganization of the cell cytoskeleton is the basis for the development of cancer progression [18]. The change of microstructure and morphology is one of the major core features of tumor malignancy, and it includes the dramatic rearrangement of cytoskeletal networks, polymerized actin/actomyosin contractility (both contribute to the enhanced motility of cancer cells) [19,20]. Microfilaments and microtubules are critical cytoskeletal-related microstructures (pseudopodia and filopodia) and are closely associated with cell motility and cancer cell malignancy. The polymerization of F-actin and β-tubulin is essential for the formation of microfilaments and microtubules [21].

Our study was designed to investigate the roles of LEF1 in regulating the cell behaviors in colonic adenocarcinoma cells (caco2) using the RNAi method, and analyze the effects of LEF1 expression on the proliferation, motility, and apoptosis of caco2 cells, as well as on the development of the motility relevant microstructures, such as pseudopodia, filopodia, and the polymerization of microfilaments and microtubules. The possible signaling pathway in the process was also explored and discussed.

## 2. Results

### 2.1. LEF1 Was Excessively Expressed in Colonic Adenocarcinoma Tissues and Enhanced Colonic Adenocarcinoma Progression

To demonstrate the potential functions of LEF1 in colonic adenocarcinoma, we used IF and GEPIA to assess LEF1 expression levels between tumor tissues and paired normal tissues. LEF1 expression was distinctly (Figure 1A) and significantly (Figure 1B,C) upregulated in colonic adenocarcinoma samples compared to normal colonic tissues. Kaplan-Meier curves showed that LEF1 expression was negatively correlated with survival rates (Figure 1D,E). The results suggest that LEF1 might play a carcinogenic role in colonic adenocarcinoma.

### 2.2. LEF1 Expression Was Efficiently Inhibited by Synthesized shRNA in Caco2 Cells

The LEF1 protein expressions highly increased in caco2 cells (*p* < 0.01) (Appendix A). To select the most effective shRNA of LEF1, three shRNA constructs targeting LEF1 mRNA were respectively transfected into caco2 cells, and shRNAscr was used as a control. The results showed a high transfection efficiency (Figure 2A). The inhibition efficiency was verified by Western blot analysis (Figure 2B). The results showed that three shRNAs suppressed the LEF1 expression efficiently, and LEF1-shRNA3 was most effective (*p* < 0.01).

### 2.3. Downregulation of LEF1 Inhibited Cell Proliferation and Motility and Induced Apoptosis

To elucidate the function of LEF1 in caco2 cells, we used LEF1-shRNA3 to inhibit endogenous LEF1 expression. The results of the MTT (Figure 3A) and colony formation assays (Figure 3B,C) showed that cell growth was significantly suppressed in the shRNA3 group compared with the WT and shRNAscr groups (*p* < 0.01).

Active motility is an important malignant phenotype of cancer cells. The effect of LEF1 inhibition on cell motility was evaluated by wound healing and Transwell assays. At 24 and 48 h following wound creation, cells in the shRNA3 group migrated significantly slower than those in the WT and shRNAscr groups (*p* < 0.01) (Figure 3D,E). The results of the Transwell assay (Figure 3F,G) showed that cells in the shRNA3 group had weaker motility than cells in the WT and shRNAscr groups (*p* < 0.01).

The loss of mitochondrial transmembrane potential is a marker of early apoptosis. MitoScene 633 staining showed that the inhibition of LEF1 potentially caused the mitochondrial membrane potential decrease in caco2 cells (Figure 3H). DNA damage is a trigger of apoptosis and we assessed the extent of DNA damage following different treatments. As shown in Figure 3I, cells in the shRNA3 group exhibited some degree of cellular retraction, while highly condensed chromatin, obvious cellular shrinkage, and abnormal shape were observed.

### 2.4. LEF1 Inhibition Remodeled Motility-Associated Microstructures

Assembly and disassembly of the F-actin bundle may significantly affect the motility, invasion, and metastasis of cancer cells [22]. The results (Figure 4A) showed that F-actin was mainly bundled along the inner surface of the cell membrane in the WT and shRNAscr groups, while in the shRNA3 group, the F-actin bundle was inhibited and disintegrated. These results indicated that the downregulation of LEF1 inhibited the polymerization of F-actin in caco2 cells. Microtubules are the critical structures to maintain cell proliferation and motility and are closely associated with the maintenance of cancer cell malignancy. The results (Figure 4B) indicate that the polymerization of β-tubulin was distinctly inhibited in the shRNA3 group. The nuclear envelope serves as an important part of the malignancy maintenance of cancer cells, and the Lamin family is an important component of the nuclear envelope [23]. Lamin B is the type V intermediate filament that is closely associated with cell proliferation [24]. Our results show that the inhibition of LEF1 visibly caused the Lamin B1 expression to be reduced (Figure 4C).

Lamellipodia and filopodia are key structures that maintain cancer cell motility and are closely related to cancer invasion and metastasis. The results (Figure 4D,E) showed that the development of cell lamellipodia and filopodia in the shRNA3 group were distinctly inhibited. Particularly, the filopodia were localized as discrete peripheral fibers-thin clusters in the interfered group compared to longer, denser, and continuous dorsal stress fibers in control groups. Most cellular events, such as metabolic pathways, including glycolysis and cell division, occur in the cytoplasm. The results (Figure 4F) showed that the cytosol had a bubble-like morphology in the shRNA3 group. The bubble-like morphology suggests that some native components in the cytosol were lost, which may cause an abnormal state of cellular activities.

### 2.5. Downregulation of LEF1 Suppressed the Expression of EMT Relevant Genes in Caco2 Cells

The effect of LEF1 inhibition on EMT was evaluated by Western blot analysis. The results (Figure 5) indicated that the expression level of E-cadherin significantly increased, while the vimentin and snail expression levels significantly decreased in the shRNA3 group (*p* < 0.01).

## 3. Discussion

LEF1 is generally excessively expressed in malignant tumors and may play a role in tumor growth and metastasis [5]. LEF1 knockdown in glioblastoma multiforme cells inhibits invasion, migration, proliferation, and the self-renewal potential of stem-like cells [25]. Myc induces the expression of LEF1 to activate the Wnt pathway in colon cancer [17]. LSD1 (lysine-specific demethylase 1) promotes bladder cancer progression by upregulating LEF1 and enhancing EMT [9]. LEF1 expression may contribute to cancer development [26,27,28], but there is a lack of evidence to support malignant phenotype changes, especially motility-associated microstructure changes, such as remodeling of lamellipodia/filopodia based on F-actin/β-tubulin polymerization.

The polymerization of F-actin and β-tubulin is essential for the formation of microfilaments and microtubules. These are the crucial cytoskeletal related microstructures (lamellipodia and filopodia) and are closely associated with cell motility and cancer malignancy [19]. The rearrangement of cytoskeletal networks contributes to the enhanced motility of cancer cells [21]. In this study, the changed cytoskeletal structures caused by LEF1 knockdown are probably related to the inhibition of cell motility. The analysis of F-actin and β-tubulin showed that downregulation of LEF1 inhibited the polymerization of F-actin and β-tubulin (Figure 4A,B). These results imply that knockdown of LEF1 inhibits microfilaments/microtubules assembly for microtentacle formation in caco2 cells, sustaining their proliferation and metastatic potential. Our results revealed that downregulation of LEF1 inhibited the development of lamellipodia/filopodia in caco2 cells (Figure 4D,E). These findings indicate that LEF1 expression is important for maintaining enhanced cell motility for the invasion and metastasis of caco2 cells, and it may be crucial to the malignancy of caco2 cells.

Strong viability and growth are essential malignant phenotypes of cancer cells. Mutations that lead to constitutive activation of the Wnt pathway drive colorectal cancer [17]. Activation of the Wnt pathway ultimately leads to the nuclear translocation of β-catenin, which binds to TCF/LEF factors to promote transcription of genes required for growth [5]. LEF1 is a transcription factor involved in the Wnt/β-catenin signaling pathway and is implicated in oncogenesis and progression of cancer [6]. LEF1 has been reported as an oncogene in tumors and may play a role in cancer invasion and metastasis [5]. In this study, we constructed LEF1 shRNA expression plasmids to estimate the roles of LEF1 in regulating the cell behaviors in colonic adenocarcinoma cells. The results demonstrated that downregulation of LEF1 inhibited the proliferation, motility, and induced apoptosis of caco2 cells (Figure 3). Lamin B1 deficiency alters the chromatin distribution and chromonema location in the nucleus in tumorigenesis [29]. Our results indicate that downregulated LEF1 impairs Lamin B1 expression and Lamin B1 plays a role in the oncogenesis and development of colonic adenocarcinoma (Figure 4C).

EMT is known to play an important role in tumorigenesis and metastasis. Cells undergoing EMT display decreased expression level of epithelial genes [such as E-cadherin, ZO-1, and occludin) and increased expression level of mesenchymal genes (such as N-cadherin, vimentin, and fibronectin) [30]. Several key signaling pathways, including TGFβ, Wnt, Notch, and Hedgehog, are known to be involved in EMT [31]. The activation of Wnt signaling inhibits the destruction complex containing glycogen synthase kinase 3 beta (GSK-3β) through Disheveled (DSH), facilitating β-catenin to enter the nucleus and activate the snail transcription [32]. LEF1 belongs to the TCF/LEF family of transcription factors and plays the role of nuclear effector in the Wnt/β-catenin signaling pathway [6,7]. We suspect that LEF1 is closely related to EMT. Our results revealed that E-cadherin expression level was increased, and the expression levels of vimentin and snail were suppressed by LEF1 inhibition in caco2 cells (Figure 5).

Our results described above support that LEF1 may play important roles during the oncogenesis and development of colonic adenocarcinoma. LEF-1 is one of the DNA-binding transcription factors in the Wnt signaling pathway, recruiting β-catenin to the transcription of Wnt target genes. Wnt signal transduction plays a key role during the oncogenesis and development of cancers, especially cancers from the digestive system [33]. Wnt signal was reported to take a crossing talk with Notch signal transduction [34,35,36]. Notch signal is aberrantly activated in various human cancers [37,38], and its inhibition can decrease the proliferation and induce apoptosis of colon cancer cells [39]. VEGF-A/Notch signaling regulates the formation of functional podosomes of endothelial cells [40]. Combining other reports with our results in the current study, LEF1 may enhance the malignancy of colonic adenocarcinoma cells intermediated by the activated Wnt/Notch signal crossing talk. The roles LEF1 plays in the merged signal regulation is schematically shown in Figure 6.

## 4. Materials and Methods

### 4.1. Colonic Adenocarcinoma Tissues

Surgical specimens were obtained from Xijing Hospital (Xi’an, China). The trial was approved by the Human Research Ethics Committee of Shaanxi Normal University (Project ID: 2021-061; Approval Date: 9 March 2021).

### 4.2. Immunofluorescence (IF)

Tissue specimens were sectioned at a thickness of 4 μm. The sections were deparaffinized with xylene and hydrated using a graded series of alcohol, and antigen retrieval and blocking were performed. The sections were incubated with primary antibody LEF1 (ab53293, Abcam, diluted 1/50 dilution) at 4 °C overnight, followed by incubation with a secondary antibody. The sections were incubated with 4’,6-diamidino-2-phenylindole (DAPI, Roche, Frankfurt, Germany) for 10 min for nuclear staining. Finally, the sections were imaged under a laser scanning confocal microscope (TCS SP5, Leica, Jen, Germany).

### 4.3. GEPIA Data Collection

Publicly available colonic adenocarcinoma data were obtained from the Gene Expression Profiling Interactive Analysis (GEPIA) database (http://gepia.cancer-pku.cn (accessed on 2 August 2021)). mRNA expression data were obtained from 624 colonic tumors and normal samples.

### 4.4. shRNA Plasmid Construction

The shRNAs targeting LEF1 were designated as described in Table 1. The shRNA expression plasmids, pU6H1-GFP-shLEF1-1, -2, -3, and controls were constructed by Biomics Biotechnologies Co. Ltd. (Nantong, China). Consequently, the pU6H1-GFP-shLEF1-3 (shRNA3) plasmid was used in subsequent experiments due to its efficiency.

### 4.5. Cell Culture and Transfection

The caco2 human colonic adenocarcinoma cell line was purchased from ATCC (Rockville, MD, USA) and cultured in DMEM medium containing 10% fetal bovine serum (FBS) at 37 °C with 5% CO_2_. Cells in the exponential phase were used for each experiment. The cells were transfected with S-TranG (Aosheng, Nanjing, China) following the manufacturer’s instructions.

### 4.6. Western Blot

Total proteins were extracted from transfected cells using RIPA lysis buffer containing protease inhibitors (Roche, Indianapolis, IN, USA). The protein concentration of the lysate was quantified using a bicinchoninic acid protein assay (BCA). Equal amounts of proteins were separated on a 7–10% SDS-PAGE gel and subsequently transferred to a methanol-activated NC membrane (Whatman, Dassel, Germany). The membranes were blocked with 5% non-fat milk in Tris-buffered saline with Tween-20 (TBST) for 1 h at room temperature and then incubated overnight with the following antibodies: LEF1 (ab53293, Abcam, 1/5000 dilution), E-cadherin (Cell Signaling Technology, 1/1000 dilution), vimentin (GTX100619, GeneTex, 1/5000 dilution), Snail (GTX100754, GeneTex, 1/500 dilution) and GAPDH (ab181602, Abcam, 1/10,000 dilution). GAPDH was used as a loading control. Then, the membranes were incubated with HRP-conjugated anti-rabbit secondary antibodies (Biosynthesis Biotech, Beijing, China). Next, the membranes were imaged using an ECL Kit (Pierce, Shanghai, China) following the manufacturer’s instructions.

### 4.7. MTT Assay

Tumor cells were seeded into 96-well plates at a density of 5000 cells/well in 200 μL/well complete medium. After transfection, MTT (3-(4,5-dimethyl-2-thiazolyl)-2,5-diphenyl-2-H-tetrazolium bromide) working buffer was added to the wells. The cells were incubated at 37 °C for 5 h, and the supernatant was discarded and replaced with 150 μL dimethyl sulfoxide (DMSO) to solubilize the converted dye. Cell viability was analyzed at a wavelength of 562 nm on an ELISA reader (Bio-Tek ELX800, BioTek Instruments, Inc., Winooski, VT, USA). DMSO-treated cells were used as a control.

### 4.8. Colony Formation Assay

At 48 h post-transfection, cells were harvested and reseeded in 6-well plates at a density of 1000 cells/well. After 2 weeks of growth, colonies were stained with 0.1% crystal violet, counted, and normalized to the control group.

### 4.9. Wound Healing Assay

Cell motility was assessed using the scratch wound healing method. Transfected cells were scratched using a tip after the cells grew as a single layer with 90% confluence. Using the area of the scratch as the cell motility parameter, the results were quantified using the online software ImageJ.

### 4.10. Transwell Assay

The cell culture and transfection methods were the same as the abovementioned methods. Cells were transferred into the top chamber (2 × 10^4^ cells/well) containing 600 μL DMEM medium without FBS, and the bottom chamber was filled with DMEM medium supplemented with 10% FBS. After incubation for 16 h, non-migrated cells were removed from the upper surface of the top section by scraping. Migrated cells attached on the other side of the top section of the membrane were stained with 0.1% crystal violet, imaged, and counted (five random fields of view per well) using an inverted microscope (Leica, Jena, Germany). The total number of cells attached on the other side of the top section of the membrane was used to evaluate the motility of the cells with different treatments.

### 4.11. Apoptotic Morphology

Cells were cultured and transfected using the abovementioned procedures. At 48 h post-transfection, the cells were incubated in DMEM with 100 nM MitoScene 633 (US Everbright Inc., Suzhou, China) or DAPI diluted in blocking buffer (1:500 dilution) at 37 °C for 30 min. The supernatant was discarded and replaced with PBS. Finally, the cells were imaged under an inverted fluorescence microscope (Leica, Jena, Germany).

### 4.12. Staining of the Microstructure-Associated Proteins

The methods for cell culture on a glass cover slip and transfection were the same as the abovementioned methods. The cover slips were incubated in 2% BSA for 1 h and then incubated with rhodamine-labeled phalloidin (1:200 dilution) and rabbit anti-human β-tubulin (1:50 dilution) or Lamin B1 (1:50 dilution) overnight at 4 °C. After 3 washes in PBS, the cover slips were incubated with a Cy3-labeled goat anti-rabbit antibody for β-tubulin and Lamin B1 staining and with DAPI for 10 min for nuclear staining. Finally, the cover slips were imaged under a laser scanning confocal microscope.

### 4.13. Coomassie Brilliant Blue Staining

Cells were cultured and transfected using the abovementioned procedures and fixed in 4% paraformaldehyde. The fixed cells were permeabilized with 0.1% Triton X-100 and then stained with 0.2% Coomassie brilliant blue R-250 (Dingguo, Xi’an, China) for 45 min at room temperature. The cover slips were imaged under an inverted microscope.

### 4.14. Observation of the Microstructural Changes

Cells were cultured on glass cover slips for 72 h after transfection. The cover slips were washed using PBS, fixed in 2.5% glutaraldehyde (Sigma, St. Louis, NY, USA) for 1 h at 4 °C and dehydrated in a gradient of diluted ethanol solutions (30, 50, 70, 80, 90, 95, and 100%). Finally, the cover slips were imaged under a scanning electron microscope (Quanta2000, Philips-FEI, Hillsboro, OR, USA) and a transmission electron microscope (JEM-2100, Tokyo, Japan).

### 4.15. Statistical Analysis

SPSS22.0 statistics software was used for all statistical analyses. Each grouped experiment was repeated 3 times and the averaged data from each group were compared with the other groups. The data are presented as the mean ± SD. Statistical analyses were carried out using ANOVA followed by the Dunnett’s post hoc test. Differences were considered statistically significant at *p* < 0.05.

## 5. Conclusions

In summary, our study has demonstrated that LEF1 enhances the motility of colonic adenocarcinoma cells via remodeling of lamellipodia/filopodia and the polymerization of F-actin/β-tubulin. LEF1 maintains the viability and growth of colonic adenocarcinoma cells through increasing proliferation, Lamin B1 expression, and decreasing apoptosis. In addition, LEF1 is closely related to EMT. These findings further support LEF1 as a potentiator and potential therapeutic target for colonic adenocarcinoma. LEF1 can be considered a novel biomarker for the evaluation and therapy of colonic adenocarcinoma.

## Figures and Tables

**Figure 1 ijms-22-10870-f001:**
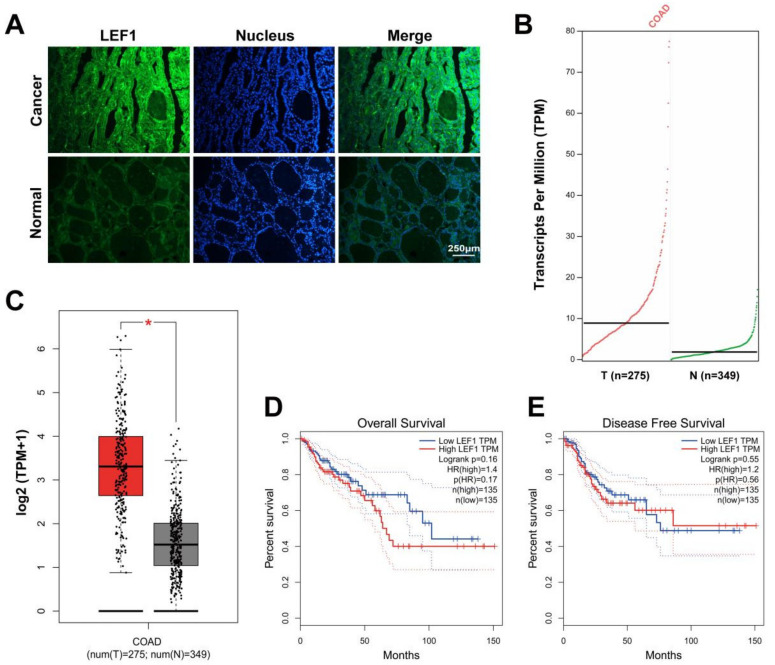
LEF1 was overexpressed in colonic adenocarcinoma tissues and reinforced the progression of colonic adenocarcinoma. (**A**): The expression level of LEF1 in colonic adenocarcinoma tissues was analyzed by IF staining (scale bar = 250 μm). (**B**,**C**): The LEF1 mRNA expression level in tumor tissues and paired normal tissues (* *p* < 0.05). (**D**,**E**): Kaplan-Meier curves showing the 10-year overall survival rate and disease free survival rate in patients with high LEF1 expression (*n* = 135) and low LEF1 expression (*n* = 135). The data were analyzed by one-way ANOVA.

**Figure 2 ijms-22-10870-f002:**
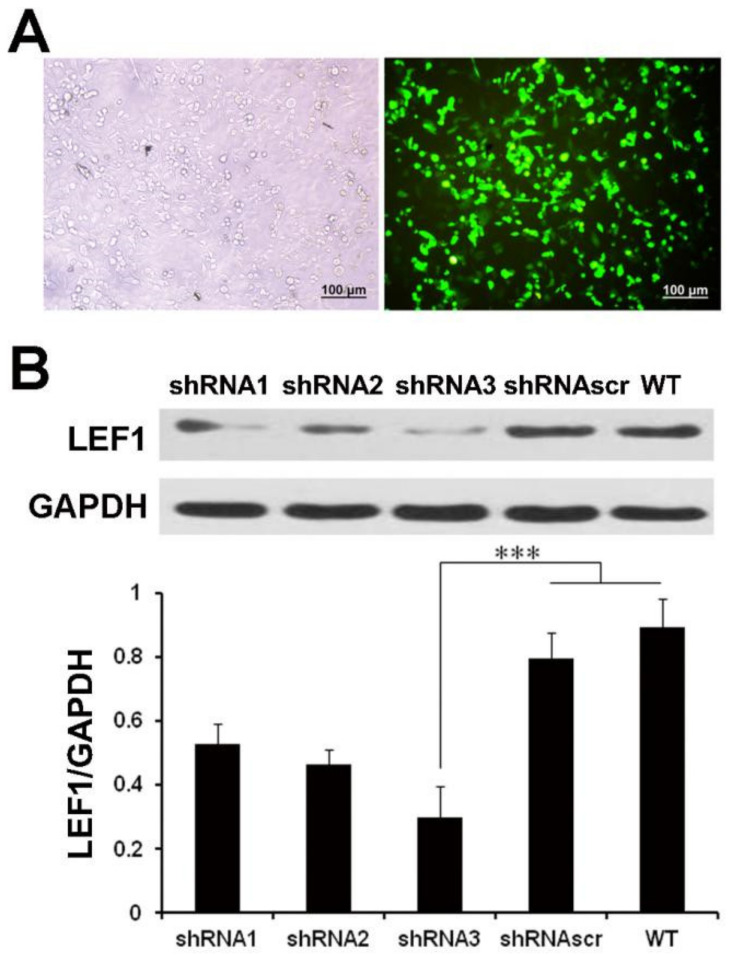
LEF1 expression was efficiently inhibited in caco2 cells. (**A**): The transfection efficiency of shRNAscr detected using an inverted fluorescence microscope at 48 h post-transfection in caco2 cells (scale bar = 100 μm). (**B**): Western blot analysis revealed that the LEF1 expression levels of the shRNA1, shRNA2 and shRNA3 groups were significantly inhibited compared with the WT and shRNAscr groups. *** *p* < 0.01.

**Figure 3 ijms-22-10870-f003:**
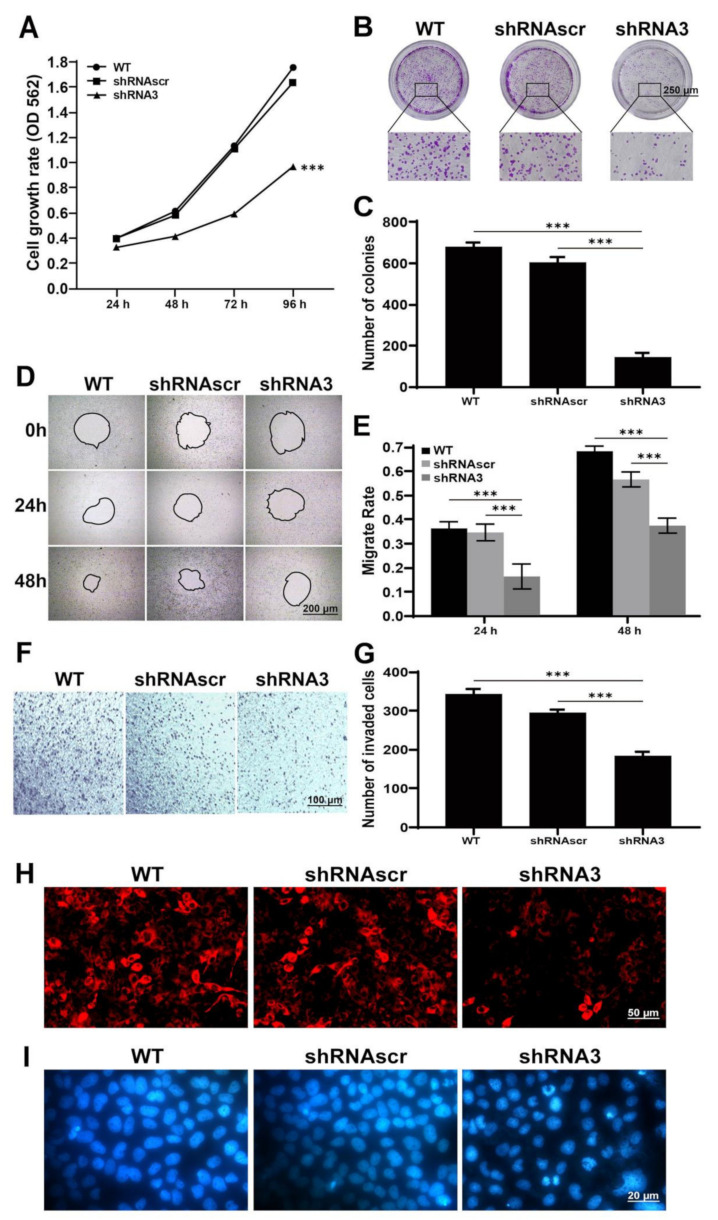
Knockdown of LEF1 inhibited cell proliferation and motility and induced apoptosis in caco2 cells. (**A**): Cell viability determined by an MTT assay at 24, 48, 72, and 96 h after transfection. (**B**,**C**): The growth of caco2 cells analyzed by a colony formation assay post-transfection (scale bar = 250 μm). (**D**,**E**): Wound healing assay at 0, 24, and 48 h after scratching (scale bar = 200 μm). (**F**,**G**): Transwell assay at 60 h after transfection (scale bar = 100 μm). (**H**): Mitochondrial detection by MitoScene 633 staining (scale bar = 50 μm). (**I**): Nuclear detection by DAPI staining (scale bar = 20 μm). *** *p* < 0.01.

**Figure 4 ijms-22-10870-f004:**
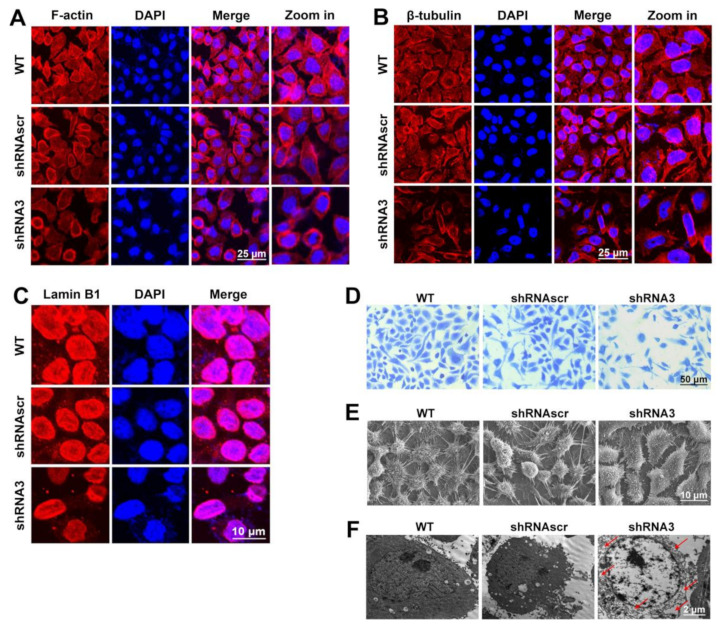
Cell microstructures were remodeled by the inhibition of LEF1 in caco2 cells. (**A**,**B**): Immunofluorescence analysis revealed that F-actin and β-tubulin bundles were distinctly disassembled in the shRNA3 group (scale bar = 25 μm). (**C**): Immunofluorescence analysis revealed that Lamin B1 expression was visibly decreased in the shRNA3 group (scale bar = 10 μm). (**D**): Coomassie brilliant blue staining (scale bar = 50 μm). (**E**): Scanning electron microscopy images (scale bar = 10 μm). (**F**): Transmission electron microscopy images (scale bar = 2 μm).

**Figure 5 ijms-22-10870-f005:**
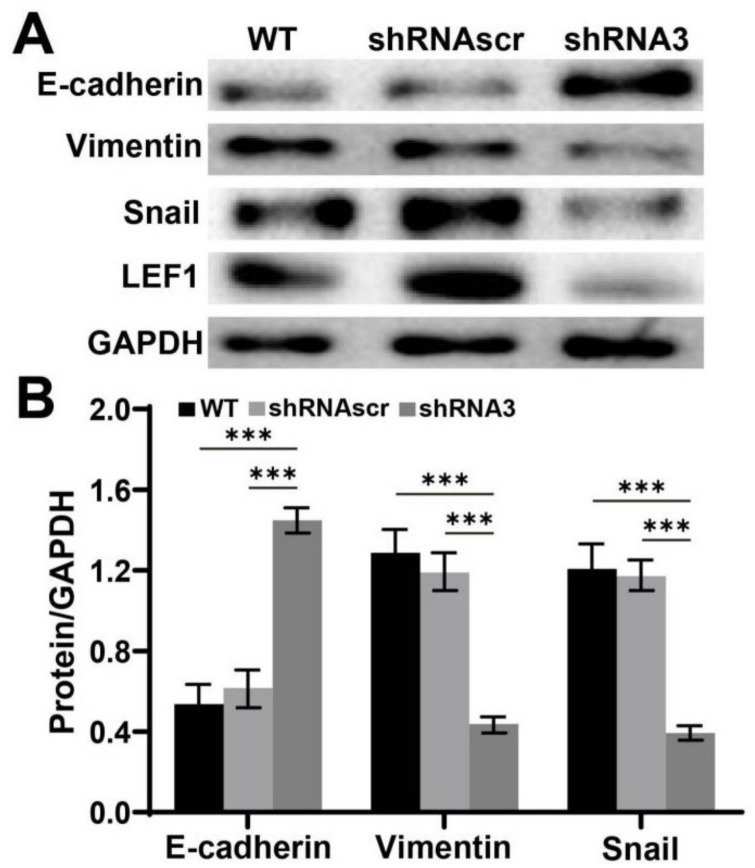
Knockdown of LEF1 inhibited the expression of EMT-relevant genes in caco2 cells. (**A**,**B**): EMT-associated gene expression analysis by Western blot. *** *p* < 0.01.

**Figure 6 ijms-22-10870-f006:**
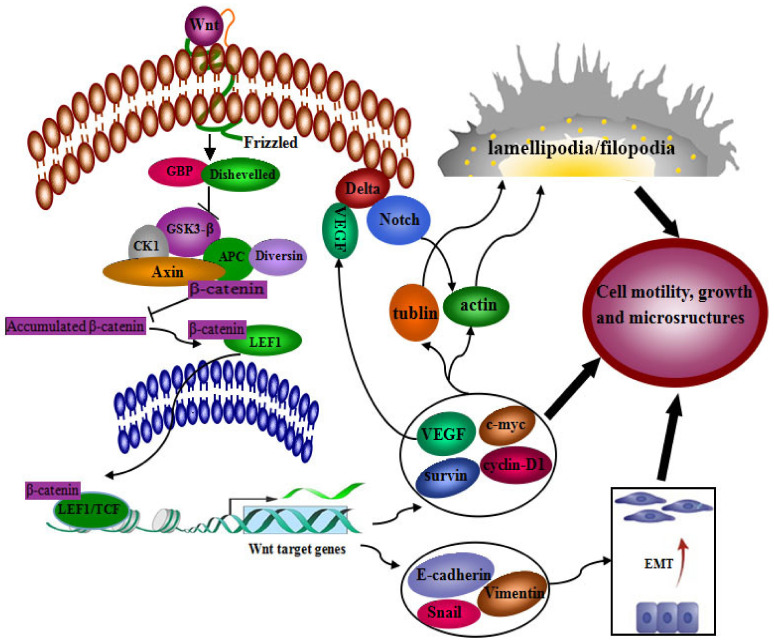
The signaling pathway of LEF1 in tumor progression.

**Table 1 ijms-22-10870-t001:** Sequences of shRNAs targeting LEF1.

shRNAs	Sequences (5′-3′)
shRNA1	GCGATTTAGCTGACATCAA
shRNA2	AGATGTCAACTCCAAACAA
shRNA3	GTTGCTGAGTGTACTCTAA
shRNAscr	TTCTCCGAACGTGTCACGT

## Data Availability

The data presented in this study are available on request from the corresponding author.

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
