# Peer review of "LEF1 Enhances the Progression of Colonic Adenocarcinoma via Remodeling the Cell Motility Associated Structures"

_ijms, 2021, doi:10.3390/ijms221910870_

Round 1

Reviewer 1 Report

In this paper, Xiao et al. examined the effects of LEF1 knockdown in caco2 cells using shRNA approach. The knockdown inhibited migration and EMT-associated molecule expression in caco2 cells. The experimental procedures are sound and the results are relatively clear. However, there still remain some uncertainties in this manuscript as shown below:

  1. The resolution of the picture shown in Fig. 1A should be improved.
  2. The resolution of the pictures shown in 4A and 4B should also be improved. Higher magnification of pictures should be used in these figures to visualize cytoskeleton more clearly. Also, what is the “bubble-like morphology” in Fig. 4F? Please explain it by marking the figure and by explaining in more detail in the text, respectively.
  3. Why was the cytoskeletal organization of the caco2 cells changed by LEF1 knockdown? Similarly, what is the reason for the reduction of proliferation and migration of the LEF1-knockdown cells compared with the WT cells? Please explain the possible mechanisms in the text.
  4. One may argue that the results shown in this paper might be applied only to caco2 cells, because the authors used only caco2 cells in their knockdown experiments. How can the authors respond to such criticism? Please explain in the text.

Author Response

We uploaded the response as a PDF file.

Reviewer 2 Report

The manuscript "LEF1 enhances the progression of colonic adenocarcinoma via remodeling the cell motility associated structures" by Xiao et al, provides evidence on the role of LEF1 in the progression of colon adenocarcinoma.

The conclusions are supported by the presented results. The study design is appropriate and the presented mechanism is clearly conceptualized.

I have detected some spelling and grammatical errors. I believe that the manuscript should be revised to avoid such errors. Apart from that the manuscript is a well-conceived and well-presented study and I would like to congratulate the authors for their effort.

Author Response

Response: Thanks to the reviewer for your time on our submission. We have corrected some English errors and revised a final linguistic revision.

Round 2

Reviewer 1 Report

The authors properly responded to my concerns.